# Blood pressure and cognitive function across the eighth decade: a prospective study of the Lothian Birth Cohort of 1936

Drew Altschul  ,[1] John Starr,[2] Ian Deary[1]

Died 8 December, 2018

[1]Department of Psychology, University of Edinburgh, Edinburgh, UK
[2]Geriatric Medicine, Royal Victoria Hospital, Edinburgh, UK

**Correspondence to**
Dr Drew Altschul;
drew.altschul@ed.ac.uk

## ABSTRACT

**Objectives** We investigated the associations among blood pressure and cognitive functions across the eighth decade, while accounting for antihypertensive medication and lifetime stability in cognitive function.

**Design** Prospective cohort study.

**Setting** This study used data from the Lothian Birth Cohort 1936 (LBC1936) study, which recruited participants living in the Lothian region of Scotland when aged 70 years, most of whom had completed an intelligence test at age 11 years.

**Participants** 1091 members of the LBC1936 with assessments of cognitive ability in childhood and older adulthood, and blood pressure measurements in older adulthood.

**Primary and secondary outcome measures** Participants were followed up at ages 70, 73, 76 and 79, and latent growth curve models and linear mixed models were used to analyse both cognitive functions and blood pressure as primary outcomes.

**Results** Blood pressure followed a quadratic trajectory in the eighth decade: on average blood pressure rose in the first waves and subsequently fell. Intercepts and trajectories were not associated between blood pressure and cognitive functions. Women with higher early-life cognitive function generally had lower blood pressure during the eighth decade. Being prescribed antihypertensive medication was associated with lower blood pressure, but not with better cognitive function.

**Conclusions** Our findings indicate that women with higher early-life cognitive function had lower later-life blood pressure. However, we did not find support for the hypothesis that rises in blood pressure and worse cognitive decline are associated with one another in the eighth decade.

## Strengths and limitations of this study

► This study used direct blood pressure measurements to model a continuous blood pressure score, as well as antihypertensive medication data, which were used to adjust blood pressure measurements.
► This study had comprehensive tests of cognitive ability measured in both childhood and old age which allowed us to investigate whether childhood and old age cognitive ability are distinctly related to blood pressure.
► Latent growth curve modelling allowed us to evaluate whether changes in either blood pressure or cognitive functions have downstream associations with one another.
► Larger samples and longer follow-up times, both from earlier and later in life, are needed to understand the long-term relationships among blood pressure, antihypertensive treatment and cognitive functions.

with lower well-being, higher morbidity and mortality and, as cognitive function worsens, the clinical conditions of mild cognitive impairment and dementia can develop.[5] Some of hypertension's negative impacts on cognitive function have likely causal pathways: hypertension disrupts cerebral blood vessel structure and function, and is associated with stroke in relevant white matter regions.[6] Deeper study of these relationships is warranted as hypertension, age-related cognitive decline and dementia are on the rise around the world.[6 7]

A major objective of modern gerontology is to understand how age-related cognitive changes systematically differ between individuals and groups. Typically, hypertension is thought of as a risk factor for later life cognitive decline. In some samples of older people, having hypertension is associated with lower cognitive functioning and faster decline.[6 8 9] However, there is also evidence for the relationship operating in the opposite direction,

## INTRODUCTION

Hypertension is a major issue in middle and older age populations.[1 2] Hypertension has been consistently linked to cardiovascular diseases such as coronary artery disease (CAD), heart failure and stroke.[1 3] It is also a risk factor for neurocognitive conditions such as age-related cognitive decline, vascular dementia and possibly Alzheimer's disease.[4] Accelerated cognitive decline is associated

that is, that higher cognitive function from earlier in life is associated with having lower risk of developing hypertension[10] and experiencing hypertension-related stroke and coronary artery events later in life.[11] These latter findings are part of a field known as cognitive epidemiology, which has found that higher cognitive function in early life is associated with lower risk of a number of physical and mental ailments later in life.[12–16] Reality is usually more complex than can be captured by a unidirectional association, as may be the case for hypertension and cognitive decline. Lower blood flow to the brain can result from hypotension, which can gradually lead to brain damage, damage that might cause further blood pressure dysregulation and subsequently worse cerebral blood flow issues.[17]

Men are more likely to develop cardiovascular conditions than women,[18] a reason why men have been the subject of more intervention studies than women.[19] Nevertheless, cardiovascular disease is the leading cause of death in both women and men.[20] Some differences in hypertension are biologically based in differences between men and women, for example, through hormones and gene dosage from the sex chromosomes, where women's additional X chromosome makes more copies of X-linked genes available for transcription. These differences are consistent across different countries and ethnic groups.[19] Additionally, traditional gender roles are associated with men behaving in ways (eg, higher smoking rates) that increase their risk for physical health conditions, including hypertension.[21] Previous work on the cognitive epidemiology of hypertension, CAD and stroke found significant interactions between sex and cognitive function in youth: individuals with higher cognitive function were at lower risk for hypertension, CAD and stroke, and the associations were stronger in women.[11 22] Socioeconomic conditions are the major avenues whereby youth cognitive function is associated with circulatory conditions. Previous work has established such factors as education[11] and income[22] as potential mediators of this sex difference.

Some cognitive functions steadily decline in mean levels in older participants, while hypertension is a condition that becomes increasingly common with age and is related to cardiovascular health and cognitive impairment. In light of this, we tested two hypotheses regarding the relationships between cognitive functions and blood pressure in the present study. First, we hypothesised that the association between higher cognitive function at age 11 years and lower blood pressure in the eighth decade of life is stronger in women than it is in men.[11] Second, we tested the hypothesis that cognitive function and blood pressure will be reciprocally associated with each other across the eighth decade. We were able to test these hypotheses using multiwave data from the Lothian Birth Cohort 1936 (LBC1936), a narrow-age cohort of over 1000 community-dwelling people, and longitudinal modelling of four waves of data that were collected from age 70 to 79 years. LBC1936 provides cognitive function

data from age 11, as well as other control variables, such as education, which has been implicated as a mediator in the relationship between cognitive function in youth and cardiovascular risk,[23–25] and a variety of related behavioural and physical health variables.

## METHOD

### Participants

The LBC1936 is a community-dwelling sample of 1091 initially healthy individuals. All were born in 1936 and were at school in Scotland on 4 June 1947, when most took part in a group-administered intelligence test: the Moray House Test (MHT) No. 12. They were followed up in four waves of one-to-one cognitive and health testing between 2004 and 2017, at mean ages 70 (n=1091), 73 (n=866), 76 (n=697) and 79 (n=550) years. Further details on the background, recruitment, attrition and data collection procedures are available.[26 27] Participants provided written informed consent. Descriptive statistics for the individual participating in each wave of the study are presented in table 1. Study completers only are described in online supplementary table 1.

### Cognitive functions

The MHT No. 12 is a broad cognitive ability test that includes word classification, proverbs, spatial items and arithmetic. The test correlated about 0.8 at age 11 years with the Terman Merrill revision of the Stanford-Binet test, providing concurrent validity.[28]

In older age, cognitive function is known to show decline across multiple, but not all, subdomains.[29] We assessed processing speed, memory and fluid cognitive ability, all of which decline on average with age. We also assessed crystallised ability, which remains relatively stable in later life.[30] Fluid and crystallised abilities are the two divisions of intelligence theorised by Cattell and Horn[31]: fluid intelligence is the inductive ability to use reasoning to solve novel problems, and crystallised intelligence is the ability to recall and apply already-known information. There are strong correlations among the subdomains, and because of this, cognitive function can be modelled hierarchically, with a general cognitive function factor that captures overall ability. Beneath that general factor, specific subdomains capture variation beyond general cognitive function.[32] The relationships among cognitive tests and subdomains are described in the Statistical analyses section.

Cognitive functions in waves 1–4 were assessed using 14 individually administered cognitive tests at the same clinical research facility and using the same equipment and procedure for all four waves. The membership of any test within a particular subdomain was determined empirically, ultimately base on model fit from confirmatory factor analysis.[32–35] The tests are fully described and referenced in an open-access protocol article.[27] The fluid subdomain consisted of matrix reasoning and block design from the Wechsler Adult Intelligence Scale

**Table 1**  Descriptive statistics for cognitive, demographic and clinical variables

|  | Wave 1 | Wave 2 | Wave 3 | Wave 4 |
|---|---|---|---|---|
| Total | 1091 | 866 | 697 | 550 |
| Female | 543 (49.8%) | 418 (48.3%) | 337 (48.4%) | 275 (50.0%) |
| Ever had high BP | 433 (39.7%) | 425 (49.1%) | 378 (54.2%) | 317 (57.6%) |
| Ever had CVD | 268 (24.6%) | 250 (28.9%) | 236 (33.95) | 204 (37.1%) |
| Ever had stroke | 54 (4.95%) | 55 (6.35%) | 73 (10.5%) | 69 (12.5%) |
| Ever had diabetes | 91 (8.34%) | 95 (11.0%) | 82 (11.8%) | 71 (13.0%) |
| Current smoker | 125 (11.5%) | 73 (8.43%) | 44 (6.31%) | 12 (3.81%) |
| Ex-smoker | 465 (42.6%) | 378 (43.6%) | 293 (42.0%) | 232 (42.2%) |
| Average sitting systolic BP | 149.6 (19.04) | 148.7 (18.83) | 148.1 (19.31) | 145.1 (18.8) |
| Average sitting diastolic BP | 81.3 (10.14) | 78.0 (9.81) | 78.9 (10.3) | 76.8 (10.1) |
| Years of education | 10.7 (1.13) | 10.8 (1.14) | 10.8 (1.14) | 10.9 (1.18) |
| Age 11 IQ | 100.0 (14.99) | 100.7 (15.3) | 101.5 (15.3) | 101.9 (15.3) |
| Matrix reasoning* | 13.5 (5.13) | 13.2 (4.96) | 13.0 (4.91) | 12.9 (5.03) |
| Block design* | 33.8 (10.3) | 33.6 (10.1) | 32.2 (9.95) | 31.2 (9.63) |
| Spatial span* | 7.04 (1.74) | 7.06 (1.61) | 7.05 (1.59) | 6.74 (1.60) |
| NART† | 34.5 (8.15) | 34.4 (8.18) | 35.0 (8.03) | 35.6 (8.19) |
| WTAR† | 41.0 (7.17) | 41.0 (6.97) | 41.1 (7.02) | 41.6 (7.03) |
| Verbal fluency† | 42.4 (12.5) | 43.2 (12.9) | 42.9 (12.8) | 43.6 (13.3) |
| Logical memory‡ | 71.5 (18.0) | 74.3 (17.9) | 74.6 (19.2) | 72.7 (20.4) |
| VPA‡ | 26.4 (9.13) | 27.2 (9.46) | 26.4 (9.56) | 27.1 (9.55) |
| Digit span‡ | 7.7 (2.26) | 7.81 (2.29) | 7.77 (2.37) | 7.56 (2.18) |
| LNS‡ | 10.9 (3.16) | 10.9 (3.08) | 10.5 (2.99) | 10.1 (2.89) |
| Symbol search§ | 24.7 (6.39) | 24.6 (6.18) | 24.6 (6.46) | 22.7 (6.63) |
| Digit-symbol coding§ | 56.6 (12.9) | 56.4 (12.3) | 53.8 (12.9) | 51.2 (13.0) |
| Inspection time§ | 112 (11.0) | 111 (11.8) | 110 (12.5) | 107 (13.6) |
| Reaction time§ | 0.64 (0.09) | 0.65 (0.09) | 0.68 (0.10) | 0.71 (0.11) |
| MMSE | 28.8 (1.43) | 28.8 (1.42) | 28.6 (1.70) | 28.48 (2.16) |

Type 2 diabetes was defined as self-reported physician diagnosis of diabetes.
*Part of the fluid ability domain.
†Part of the crystallised ability domain.
‡Part of the memory domain.
§Part of the processing speed domain.
BP, blood pressure; CVD, cardiovascular disease; LNS, letter-number sequencing; MMSE, Mini-Mental State Examination; NART, National Adult Reading Test; VPA, Visual Paired Associates; WTAR, Wechsler Test of Adult Reading.

(WAIS),[36] and spatial span forward and backward from the Wechsler Memory Scale (WMS).[37] Processing speed was measured through symbol search and digit symbol substitution from the WAIS, plus four-choice reaction time[38] and inspection time.[39] Memory was assessed using verbal paired associates and logical memory from the WMS,[37] and the letter–number sequencing and digit span backward subtests of the WAIS.[36] Crystallised ability was measured through the National Adult Reading Test,[40] Wechsler Test of Adult Reading[41] and a phonemic verbal fluency test.[42]

### Blood pressure
Blood pressure was measured six times at each wave, three times sitting and three times standing. Blood pressure measurements were divided into systolic pressure (maximum during one heartbeat) and diastolic pressure (minimum in between two heartbeats).

In addition to working with raw, unadjusted blood pressure measurements, we also wished to study a more natural course of blood pressure development that removed the potentially confounding influence of antihypertensive medication. Adjustments for antihypertensive medications were made to both systolic and diastolic blood pressure. Fifteen points were added to systolic pressure and 10 points were added to diastolic pressure[43] if an individual was recorded as taking antihypertensive medication, such as atenolol, lisinopril, bisoprolol and so on at that wave.

## Covariates

Sex and education were both included as covariates in all models. Education was recorded as the total number of years spent in formal education. Age 11 cognitive function was also included in all models, as was the interaction between sex and age 11 function.

Several behavioural and diagnosed health variable were included. Smoking status was dummy coded with two variables, current smokers and ex-smokers, with non-smokers as the reference group. Participants were genotyped for the presence of an *APOE ε4* allele. Cardiovascular disease and stroke history were both recorded for each wave; each was coded as a binary variable, with a 1 indicating that the individual had a history of the disease. Testing dates varied slightly for every individual at every wave, so the exact age of each participant at testing was also recorded.

## Statistical analyses

All four LBC1936 Waves were analysed in latent growth curve models (LGCMs), a structural equation modelling technique that allows the user simultaneously to define and analyse multiple latent and measured variables.[44] LGCMs make use of latent variables, which are constructed from multiple indicators, that is, observed variables. Incorporating the multiple measures of systolic and diastolic blood pressure allows the latent variables in LGCMs to account for measurement error,[45] and in LGCMs blood pressure can be modelled holistically and longitudinally with level (ie, an intercept) and slope (ie, a trajectory of change) parameters.

We modelled blood pressure variables using a hierarchical 'curve of factors' model (online supplementary figure 1). At the bottom tier were individual systolic and diastolic readings (only sitting readings were used for LGCMs), grouped by wave. For each grouping, both a systolic and diastolic latent variable were specified, creating four latent variables each for systolic and diastolic pressure, one each per wave. From these latent systolic and diastolic variables, latent variables modelling growth was specified, in much the same way as with cognitive function. To model overall blood pressure level, each of the eight lower order latent variables contributed its unit loaded value (ie, x1). For blood pressure, the best fit models were those that included both linear and quadratic slope latent variables. To model linear slope of blood pressure, linearly increasing loadings were given as the average age at a given wave progressed (x0, x2.96, x6.72, x9.79). To model quadratic slope, quadratically increasing loadings were assigned (x0, x8.76, x45.16, x95.84).

We modelled cognitive function using a slightly different hierarchical 'factor of curves' model, previously established with these data.[32] For each cognitive test, we modelled a level (essentially the age 70 baseline) and a linear slope (the change between age 70 and age 79, taking all four measurement occasions into account) with the same loadings as with the age-respective blood pressure variables. For each cognitive domain (see the Cognitive functions section), a latent level and linear slope variable was correspondingly composed of the individual tests' latent level and latent slope variables. At the top of this cognitive hierarchy, latent levels and slopes for general cognitive function were formed from the level and slope variables of each cognitive domain. Similar models have been used and diagrams presented elsewhere[32 34] and a structural diagram to illustrate these measured and latent variables and the associations among them is presented in online supplementary figure 2.

Put simply, BP level is analogous to BP at the study's beginning (age 70 years), BP slope is analogous to the linear magnitude at which BP increases or decreases over the following decade, and BP quadratic (slope) is analogous to the curvature of the change in BP over the following decade. Cognitive function level is analogous to cognitive function at age 70, and is known to be highly correlated with cognitive function at age 11,[32] demonstrating some stability of individual differences in cognitive function across the life course. Slopes of cognitive function represent the rate at which cognitive functions change (mostly decrease, in fact) over subsequent waves, which is otherwise known as cognitive decline.

Time-invariant covariates that applied to all waves of data (sex, age 11 cognitive function, their interaction, years of education and smoking history) were associated with the intercept and slope parameters for latent levels and slopes, including those for the domains of cognitive function, but not onto individual cognitive tests' levels or slopes. *APOE ε4* status only applied to cognitive function latent variables. Individual systolic and diastolic BP measurements were associated with time-varying covariates, that is, characteristics of individual waves: age at testing, cardiovascular disease history and stroke history. LGCM 1A included the time invariant covariates sex, age 11 cognitive functions, their interaction, and education, as well as time-varying age at date of assessment. Model 1B included these covariates as well as *APOE ε4* status and time-varying factors of smoking and disease history.

Model parameters were estimated using full-information maximum likelihood, that is, all data were used for all participants, even individuals who did not complete all waves. Standard errors were calculated using the robust Ruber-White method, and p values were computed using the Yuan-Bentler scaled test statistic.[46] The false-discovery rate correction for multiple testing was applied to the variables in each model that were not control covariates, which included the associations between the latent variables of cognitive function and its subdomains and BP, as well as associations from sex, age 11 cognitive function and their interaction. All analyses were conducted in the R programming language (V3.3.2), using the 'lavaan' package for modelling.[47]

## Patient and public involvement

LBC1936 participants were not involved in the development of any part of this study. The results will be

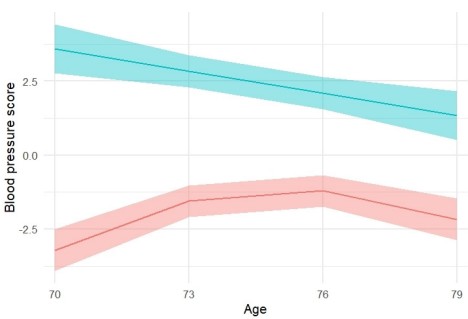

**Figure 1** Blood pressure trajectories in the eighth decade for individuals with and without hypertension diagnoses. Only study completers, those participants who were present at every wave of the study, were included in this plot. All study participants were used in the statistical analyses. Blood pressure score was derived from the latent variable that represent blood pressure in individual waves. Blood pressure score is a standardised, unitless measure of overall blood pressure magnitude; A score of 0 indicates the individual is in line with average blood pressure, and 1 would indicate that an individual was 1 SD higher blood pressure at that time. The upper green line illustrates individuals diagnosed with hypertension at any wave (n=337) and the lower red line illustrates normotensive individuals (n=204). The shaded areas represent 95% confidence regions.

disseminated to participants via a quarterly newsletter sent to LBC1936 participants.

## RESULTS
### Modelling adjusted blood pressure

The best fit model of BP adjusted for medication was a good fit for these data ($\chi^2$=2052.76, df=259, p<0.001, Comparative Fit Index (CFI)=0.931, Root Mean Square Error of Approximation (RMSEA)=0.080). The correlations between the level, linear slope and quadratic slope latent BP variables were as follows: $r_{level,slope}$=−0.466

($z$=−10.216, p<0.001), $r_{level,quad}$=0.323 ($z$=5.525, p<0.001), $r_{slope,quad}$=−0.466 ($z$=−90.460, p<0.001).

Blood pressure scores for each wave were derived from the latent variables and illustrated in figure 1. In both hypertensive and normotensive individuals, blood pressure scores were similar at each age except for a notable but unsurprising difference in magnitude: individuals with hypertension had much higher blood pressure than those who did not have hypertension. Normotensive individuals show a quadratic effect, blood pressure rose between ages 70 and 76, and then declined between age 76 and 79.

### Adjusted blood pressure and cognitive function

The best fit model of both adjusted blood pressure and cognitive function included latent variables for level and linear slope ($\chi^2$=8911.01, df=4023, p<0.001, CFI=0.928, RMSEA=0.033). Models including a quadratic factor of cognitive function change either would not converge successfully or fit very poorly and could not be trusted to produce reliable estimates for cognitive function, and thus were not included in our models. Adjusted blood pressure was best modelled with a quadratic factor of change.

Overall, we found no associations between latent variables of cognitive function with those of blood pressure in our bivariate growth curve (model 1A, figure 2 and online supplementary table 2). After correction for multiple comparisons, only three tested associations (out of 22) survived. First, a correlation between eighth decade cognitive function level and slope ($r$=0.326, SE=0.059, p<0.001) suggests that individuals with higher overall cognitive function have steeper cognitive decline over the eighth decade, likely because they have more functional ability to lose. Second, there was a regression path from age 11 cognitive function to eighth decade cognitive function

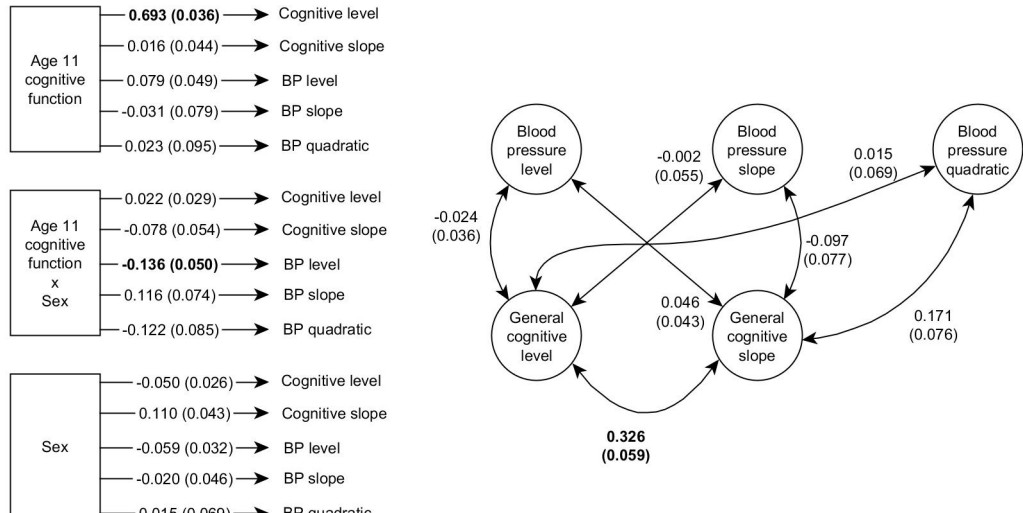

**Figure 2** Path diagram displaying the statistical tests in a latent growth curve model of cognitive function and blood pressure (BP). Circles represent latent variables and squares represent measured variables. Double arrowed lines represent correlations. Single arrowed lines (on the left side) represent regressions. Coefficients and SEs are the result of model 1A; numbers printed in bold indicate coefficients that were statistically significant after correction for multiple comparisons.

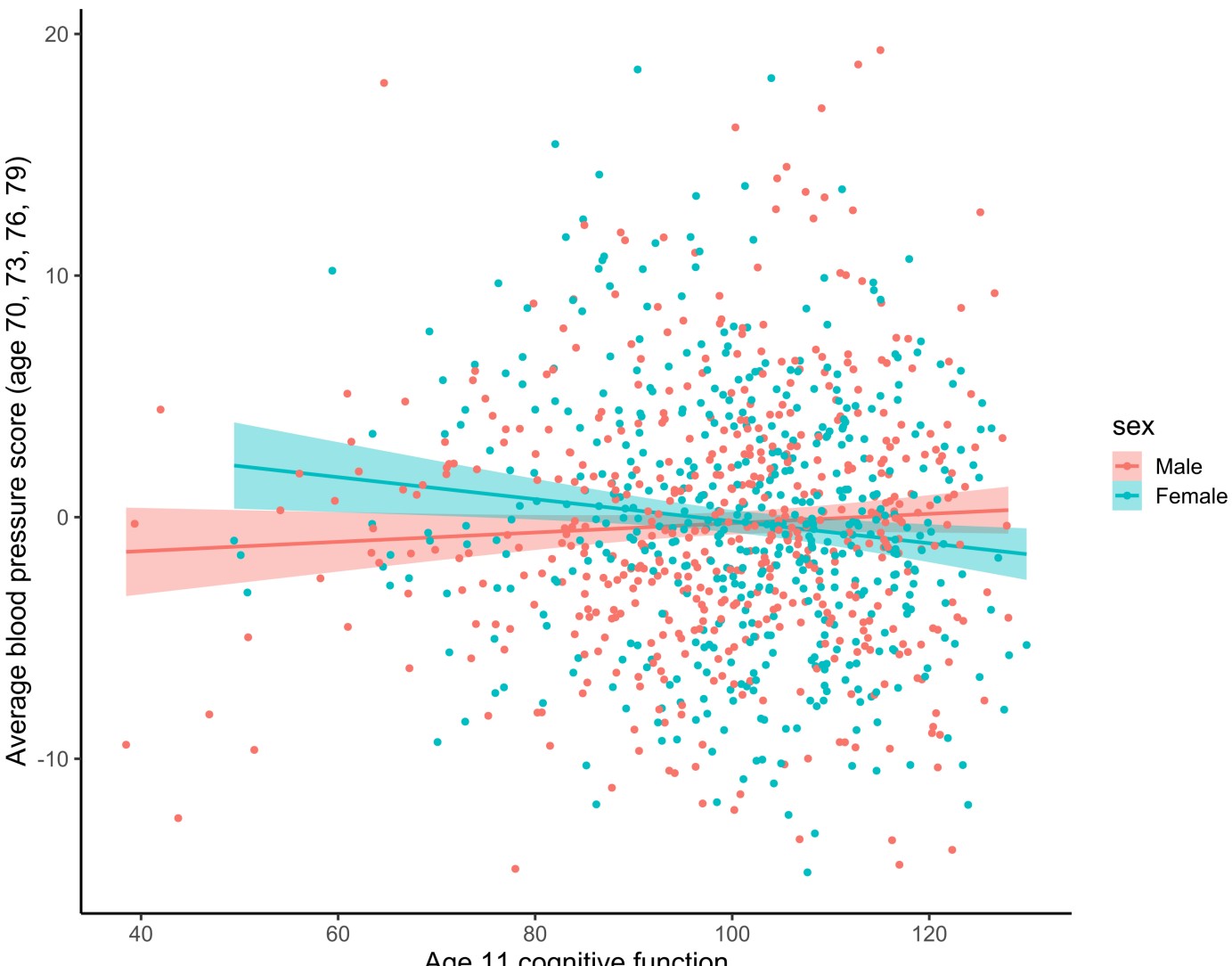

**Figure 3** Blood pressure in older age and its relationship with cognitive function at age 11 years, sex and their interaction. Blood pressure score was derived from the latent variable that represent blood pressure in individual waves. The green dots and line represent women, and the red dots and line represent men. The shaded areas represent 95% confidence regions.

level ($\beta$=0.693, SE=0.036, p<0.001), which reproduces a well-known phenomenon: cognitive function shows substantial stability of individual differences across the life course,[48] and different measures taken at different times will still be strongly associated.

Third, a significant regression path was found from the interaction of age 11 cognitive function and sex to eighth decade blood pressure level ($\beta$=−0.136, SE=0.050, p=0.047). This interaction association indicates that women with higher cognitive function at age 11 have lower blood pressure in later life, and women with lower cognitive function have higher blood pressure. The opposite was true for men: males with higher cognitive function at age 11 had higher blood pressure and vice versa (figure 3). No additional associations were found between any specific cognitive domains and blood pressure variables. All these tested associations are visualised in figure 2.

Incorporating control variables into the LGCM (model 1B, online supplementary table 3) reduced the regression weight from the sex and age 11 cognitive function interaction to blood pressure level, so that it was no longer significant ($\beta$=−0.106, SE=0.058, p=0.215). The variable that most likely caused this change in the model was smoking behaviour; specifically, being a current smoker at age 70 had an association with eighth decade blood pressure ($\beta$=−0.104, SE=0.040, post-hoc uncorrected p=0.009).

### Unadjusted blood pressure and cognitive function
We also attempted to fit bivariate LGCMs of unadjusted blood pressure and cognitive function. These models were intended to be the same as the models discussed above, but without the blood pressure variables being adjusted when individuals were recorded as taking antihypertensive medication. These models would not converge, and we were unable to progress further with this analysis. In order to investigate the influence of medication further,

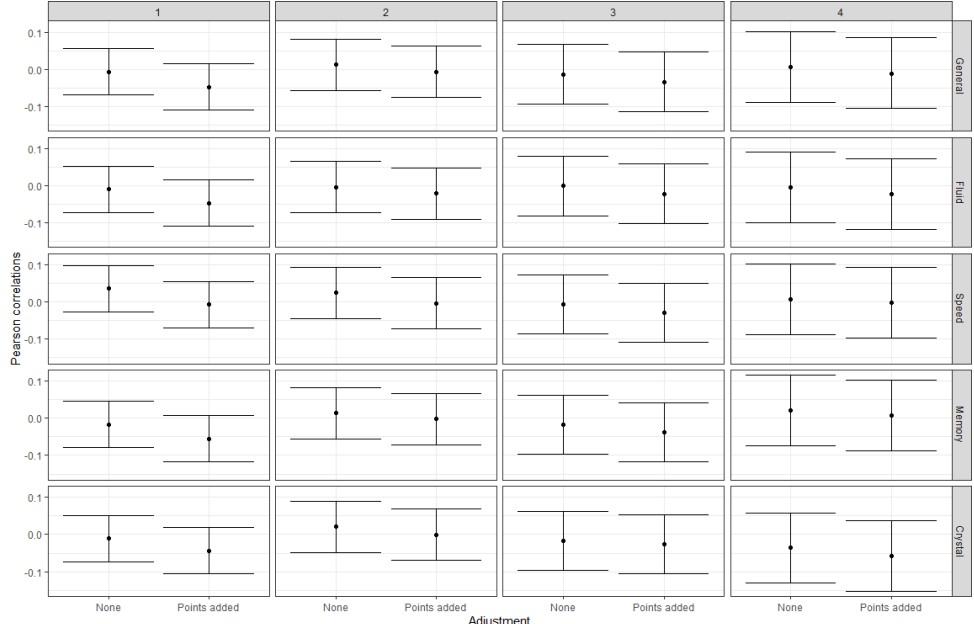

**Figure 4** Pearson correlations between general and subdomain cognitive function scores and blood pressure scores within each wave of the Lothian Birth Cohort. Wave is indicated across the top of each column; the left side of each pane represents correlations among data without adjustment for antihypertensive treatment, and the right side of each pane represents correlations among data with adjustment for antihypertensive treatment. Error bars are 95% CIs around the correlation coefficient.

we carried out cross-sectional and additional longitudinal analyses.

### Cross-sectional correlations between cognitive functions and blood pressure scores across waves

Given the quadratic trajectory of blood pressure over time in these data, and earlier evidence suggesting that the association between cognitive function and blood pressure changes depending on age,[49] we examined correlations between cognitive function, both domains and general ability, and blood pressure scores, both with and without adjustment for antihypertensive medication. We carried out these correlations within wave, that is, we looked at correlations between age 70 cognitive function and age 70 blood pressure, age 73 cognitive function and age 73, blood pressure, and so on for ages 76 and 79.

The results of these analyses are shown in figure 4 and online supplementary table 4. Correlations were similar at all time points, although all correlations were also small; no 95% CIs did not overlap with zero.

### Diagnosis and treatment effects

In an effort to understand why we found no associations between cognitive function and blood pressure in the eighth decade, we examined the role of medication in post-hoc analyses. The majority of individuals diagnosed with hypertension in the LBC1936 were prescribed medication for the condition. Following from figure 1, though individuals with hypertension would have had higher blood pressure were they not being treated, they are being treated, and their blood pressure appears to generally decline over the eighth decade.

We fit linear mixed effect models to explore the associations of diagnosis and medication with blood pressure. Using latent variables for blood pressure, we generated blood pressure scores at each wave, but, by contrast with the previous analyses, the scores were based on the raw, medication-unadjusted systolic and diastolic blood pressure measurements. Using these scores, we first fit a mixed model predicting blood pressure score from a two-way model that included the wave of the study, whether an individual had a hypertension diagnosis at that wave, and the interaction of the two (model 2A, table 2). We found a significant main effect of wave on blood pressure score ($\beta$=−0.048, SE=0.014, p<0.001) and interaction between wave and hypertension diagnosis ($\beta$=−0.111, SE=0.028, p<0.001), suggesting that individuals who are given a hypertension diagnosis have decreasing blood pressure over time.

In a second mixed effects model, we replaced hypertension diagnosis by wave with a single variable indicating if a hypertension diagnosis was given at any wave, that is, if a participant was diagnosed with hypertension during the eighth decade. This operationalisation of diagnosis may capture more information than individual wave diagnoses; for example, an individual who first reports a hypertension diagnosis at wave 3 might already have hypertension at wave 2, and even if they do not have hypertension at wave 1, their blood pressure is likely to be elevated as they will be on their way to developing clinical hypertension. By coding all this information in a single variable, we reduce the number of comparisons in the models as well as potentially improving our operationalisation. As

**Table 2** Linear mixed effects regression models of blood pressure and cognitive function scores predicted by antihypertensive treatment over time

| Variable | Std estimate | SE | Wald χ2 | P value |
|---|---|---|---|---|
| Predicting blood pressure score—model 2A | | | | |
| Wave | −0.048 | 0.014 | 11.854 | <0.001 |
| Did the individual have HT at this wave? | −0.031 | 0.023 | 0.339 | 0.560 |
| Wave × having HT | −0.111 | 0.028 | 15.370 | <0.001 |
| Predicting blood pressure score—model 2B | | | | |
| Wave | −0.086 | 0.015 | 20.929 | <0.001 |
| Did the individual have HT at any wave? | 0.170 | 0.031 | 35.162 | <0.001 |
| Wave × having HT | −0.208 | 0.032 | 43.536 | <0.001 |
| Predicting cognitive function score—model 3A | | | | |
| Wave | −0.154 | 0.007 | 490.085 | <0.001 |
| Did the individual have HT at *this* wave? | −0.008 | 0.013 | 0.552 | 0.458 |
| Age 11 cognitive function | 0.680 | 0.023 | 945.862 | <0.001 |
| Wave × having HT | 0.006 | 0.015 | 0.148 | 0.701 |
| Wave × age 11 cognitive function | −0.039 | 0.014 | 7.512 | 0.006 |
| Having HT × age 11 cognitive function | −0.037 | 0.028 | 1.751 | 0.186 |
| Wave x having HT × age 11 function | −0.003 | 0.029 | 0.009 | 0.923 |
| Predicting cognitive function score—model 3B | | | | |
| Wave | −0.157 | 0.007 | 517.451 | <0.001 |
| Did the individual have HT at *any* wave? | −0.105 | 0.029 | 13.230 | <0.001 |
| Age 11 cognitive function | 0.678 | 0.025 | 754.084 | <0.001 |
| Wave × having HT | −0.007 | 0.015 | 0.284 | 0.594 |
| Wave × age 11 cognitive function | −0.041 | 0.015 | 7.747 | 0.005 |
| Having HT × age 11 cognitive function | 0.020 | 0.054 | 0.165 | 0.685 |
| Wave × having HT × age 11 function | −0.014 | 0.029 | 0.216 | 0.642 |

HT, hypertension.

a result (model 2B, table 2), we found similar associations as in model 2A, except that we also found a main effect of being diagnosed with hypertension at any wave, indicating that individuals with a diagnosis generally had higher blood pressure as we would expect.

We fit two additional linear mixed effects models to examine associations between hypertension diagnosis and cognitive function. Because later life cognitive function is strongly associated with early life cognitive function, we included age 11 cognitive function as an independent variable in our regression models, and included all two-way and three-way interactions with wave and whether an individual was diagnosed with hypertension.

These models (models 3A and 3B, table 2) reproduced several established effects.[32 48] Age 11 cognitive function is associated with cognitive function scores in the eighth decade ($\beta$=0.680, SE=0.023, p<0.001). Cognitive function declines across successive waves ($\beta$=−0.154, SE=0.007, p<0.001). Individuals with higher cognitive function early in life have more ability to lose, so declines are greater for these individuals across time ($\beta$=−0.039, SE=0.014, p=0.006).

In the second model of this set of models (model 3B, table 2), there was also a main effect of being diagnosed with hypertension at any wave and overall cognitive function level ($\beta$=−0.105, SE=0.029, p<0.001). Having a hypertension diagnosis was associated with lower cognitive function, which could not be fully explained by cognitive function scores from earlier in life. Put another way, being diagnosed with hypertension was not associated with improved cognitive function nor could diagnosis account for the negative associations between blood pressure and cognitive decline. In previous models (table 2), being diagnosed with hypertension was associated with reduced blood pressure over time. Therefore, whereas hypertension diagnoses and antihypertensive medication appear to successfully treat high blood pressure in this cohort, they do not appear to impact associated cognitive deficiencies.

## DISCUSSION

Our results show that sex and cognitive function from early-life interact to predict blood pressure level during

the eighth decade. Women with higher cognitive function are less likely than higher cognitively functioning men to have higher blood pressure, but the opposite is also true: lower cognitive function women are more likely to have higher blood pressure than similarly functioning men (figure 3). This finding provides mixed support for our first hypothesis and the existing supporting literature,[19 22] as the effect was not robust to the inclusion of covariates. Some of this association can be explained by health behaviours and other health conditions, some of which stand out. Notably, whether one is a smoker or not appeared to explain some of the same variation as the sex by cognitive function interaction.

We did not find compelling evidence to support our second hypothesis. There were no associations between high blood pressure and cognitive function, either in level or trajectory of function, nor were any other health or sociodemographic variables associated with change in either blood pressure or cognitive function in the eighth decade. As a null result, this is a difficult finding to interpret. It may be that we did not have sufficient power to detect such associations in our sample, although nearly 40% of the sample had been diagnosed with hypertension by the initial wave at age 70. Another possibility is that cognitive function is most impacted by side effects of hypertension before the eighth decade: our findings tentatively support this, as individuals with a hypertension diagnosis tended to have lower cognitive function, below what we would expect from baseline assessments at age 11.

Another possibility is that antihypertensive medications were effective at reducing hypertensive symptoms, which includes the connections between cognitive function and hypertension. Our findings suggest that participants of this sample who are diagnosed with hypertension take their medication, and this had a significant, ameliorative effect on their raised blood pressure. However, we found no evidence that either medication or behavioural changes that might result from being diagnosed with hypertension have a positive association with subsequent cognitive function. In other words, treatment from antihypertensive medication in the eighth decade did not impact age or hypertension-related cognitive decline in this sample. These findings are consistent with large trials that have found no significant link between controlling blood pressure and dementia incidence.[50 51]

There is existing evidence for the importance of lifestyle factors in explaining the associations between cognitive function and physical health. In the National Longitudinal Study of Youth 1979, higher cognitive function women were less likely to be given a hypertension diagnosis,[22] an interaction that closely matches our initial finding in the LBC1936. However, this effect was explained by income differences. The Aberdeen Children of the 1950s cohort yielded results that were also similar to ours; specifically, associations between childhood cognitive function and both stroke and coronary artery events were present in both sexes, but stronger in women.[11] However, in their analyses, the sex by cognitive function interaction effects on stroke and CAD outcomes could be accounted for by education. In both of these cases, there were socioeconomic factors that seemed to drive the interaction between early-life cognitive function and sex, whereas we found that behaviours, specifically smoking, seemed to explain some of the interaction. These samples vary in age and location, so chronological and geographic-cultural cohort differences might explain the discrepancies.[52 53]

The present study is limited by a non-trivial proportion of missing data, particularly from individuals who died over the course of the study, or were too frail or otherwise unwilling to continue participating. Moreover, our analytic sample was more affluent than the average population, and may thus limit generalisability. The diagnoses and medications analysed in the present study were self-reported and we were not able to cross-reference these reports with any physician records. Although we took steps to treat these variables conservatively in our analyses, self-reported diagnoses of hypertension tend to have lower validity than those drawn from medical records.[54]

## CONCLUSIONS

In general, our results are consistent with previous work that has indicated that the effects of lower childhood cognitive function on hypertension are stronger and more consistent in women.[11 22] Despite there being known connections between high blood pressure and cognitive impairment, we did not find any evidence for this in the eighth decade. It appears that any cognitive lowering associated with hypertension may occur earlier in life. Future research ought to corroborate this and, in general, investigate the critical periods when hypertension might be detrimental to cognitive health.

**Acknowledgements** The LBC1936 study is funded by Age UK (Disconnected Mind project). The authors thank the Scottish Council for Research in Education for allowing access to the SMS1947. They thank the LBC1936 study participants and research team members.

**Contributors** DA discussed and planned the study and analyses, analysed the data, interpreted the data and drafted the initial manuscript. JS and ID discussed and planned the study and analyses, interpreted the data and contributed to the manuscript.

**Funding** The LBC1936 data were collected using a Research Into Ageing programme grant; this research continues as part of the Age UK-funded Disconnected Mind project. DA, JS and ID were members of The University of Edinburgh Centre for Cognitive Ageing and Cognitive Epidemiology, which was funded by the Biotechnology and Biological Sciences Research Council and Medical Research Council (MR/K026992/1). DA is also funded by an MRC Mental Health Data Pathfinder award (MC_PC_17209).

**Competing interests** None declared.

**Patient and public involvement** Patients and/or the public were not involved in the design, or conduct, or reporting, or dissemination plans of this research.

**Patient consent for publication** Not required.

**Ethics approval** Ethics permissions were obtained from the Multicentre Research Ethics Committee for Scotland (wave 1, MREC/01/0/56), the Lothian Research Ethics Committee (wave 1, LREC/2003/2/29) and the Scotland a Research Ethics Committee (waves 2–4, 07/MRE00/58).

**Provenance and peer review** Not commissioned; externally peer reviewed.

**Data availability statement** Data are available upon reasonable request. Lothian Birth Cohort 1936 data can be requested from the Lothian Birth Cohort 1936 research team, following completion of a data request application. More information can be found online (http://www.lothianbirthcohort.ed.ac.uk/content/collaboration).

**ORCID iD**
Drew Altschul http://orcid.org/0000-0001-7053-4209

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
