## [Reviewer comments · BMJ Open]

ARTICLE DETAILS

TITLE (PROVISIONAL)	Blood pressure and cognitive function across the eighth decade: a prospective study of the Lothian Birth Cohort of 1936
AUTHORS	Altschul, Drew; Starr, John; Deary, Ian

VERSION 1 - REVIEW

REVIEWER	rudi westendorp University of Copenhagen, Denmark
REVIEW RETURNED	04-Oct-2019

GENERAL COMMENTS	This is scholarly analysis of the relation between blood pressure and cognitive function across the eighth decade within the phenotypically rich Lothian Birth Cohort of 1936. Outcomes of this type of analyses are quintessential for a better understanding of cognitive decline in old age which is of interest for a great readership. The authors go to great length to understand why there were no associations between cognitive function and blood pressure in the eighth decade, and therefore examined the role of medication in post-hoc analyses. In the main analysis (line 160), adjustments for anti-hypertensive medications were made to both systolic and diastolic blood pressure. 15 points were added to systolic pressure and 10 points were added to diastolic pressure if an individual was recorded as taking antihypertensive medication, such as atenolol, lisinopril, bisoprolol, etc. at that wave. I follow this reasoning when testing the hypothesis that the association between higher cognitive function at age 11 years and lower blood pressure in the eighth decade of life is stronger in women than it is in men, as in that case participants would have a 'falsely' low recorded blood pressure. However, when the hypothesis is tested that cognitive function and blood pressure will be reciprocally associated with each other across the eighth decade, an etiologically question, statistically adding blood pressure in a prospective design when the real blood pressure is measured, is not obvious. When fitting their posthoc models, there was an inverse main effect of being diagnosed with hypertension at any wave and overall cognitive function level (line 330) suggesting that hypertension may be associated with negative cognitive change between childhood and older age, which is in line with current reasoning on the relation between longstanding hypertension and cognitive decline. However, the following reasoning 'that being diagnosed with hypertension did not have any protective associations with cognitive function across the eighth decade: it did not improve cognitive function or make up
--

	for the negative association between blood pressure and cognitive decline.’ (line 335) is rather ambiguous. Why should a history of hypertension protect against cognitive decline in old age? Please note that this question is different from the pathophysiological reasoning that higher blood pressure in old age may protect against cognitive decline, especially in those with cerebral vascular damage (due to long-lasting hypertension). My first suggestion is that in the second paragraph of the introduction (Line 86) the authors describe a third type of association between blood pressure and cognition, based on pathophysiological reasoning, for instance see https://www.ncbi.nlm.nih.gov/pubmed/25730401. From this text, ‘It has been shown that the degree of vascular damage in the systemic and cerebral circulation is linked with lower cerebral blood flow. Long-lasting cerebral hypoperfusion results in neuronal energy crisis and cell death. At the same time, damage of the brain can lead to dysregulation of blood pressure and a further decline in cerebral blood flow. Therefore, what is considered a normal or low blood pressure in individuals with cognitive impairment may not necessarily mean a well-controlled blood pressure.’ My second suggestion is to study the effect of blood pressure and cognition later in life, making full use of the follow-up design, as blood pressure is a causal determinant of cognition. For an example see https://www.ncbi.nlm.nih.gov/pubmed/19453303. In contrast, the structural equation modelling that is being used when analyzing the Lothian Birth cohort, effectively makes it a cross sectional design. The outcomes of cross-sectional analyses on blood pressure and cognition generally indicate a positive association, suggesting that high blood pressure is protective. This is, as a general conclusion unjust and all kinds of biases, confounding and selection mechanisms are at play. My third suggestion is to combine these two thoughts, i.e. allowing that there is an age and or state (i.e. a history of hypertension and or cerebral vessel disease) interaction between blood pressure and cognition. As a beginning, it may be worthwhile to split the sample in two, analysing the first half of the observation period separate from the second half. There is a strong lead in the data that there is an age and or state interaction at play as ‘Both groups show a quadratic effect, blood pressure rose between ages 70 and 76, and then declined between age 76-79 (line 249). Rudi Westendorp, Copenhagen.
--	---

REVIEWER	j gussekloo leiden university medical center
REVIEW RETURNED	25-Jan-2020

GENERAL COMMENTS	This manuscript is well written and is focused on blood pressure and cognitive function, a very interesting and actual scientific theme. However, since 1) the clinical hypothesis is not very well worked out and 2) the statistical methods and results are very difficult and extensive, I believe this manuscript fits better in a statistical journal instead of this journal for clinical doctors. In detail, I have the following extra points.
---

A. Introduction:

- Line 77: "...linked to cardiovascular diseases such as coronary artery disease (CAD) and stroke." I would suggest to add also heart failure because the link between chronic high blood pressure and heart failure is of great clinical importance (the prevalence of heart failure rises to 10% and more among people 70 years and older).
- Line 84: "Worldwide, hypertension, age related cognitive decline, and dementia are on the rise." Important fact, but it pops a little bit out of the blue at the end of the first paragraph of the introduction. Can it be streamlined into the paragraph?
- Line 88: "In some samples of older people hypertension is associated with lower cognitive functioning and faster decline." I would advise to include 'the timing' of hypertension. Do the authors mean hypertension since middle-age (or even adulthood) or development of hypertension at an older age? In pathophysiological context this can be somewhat different.
- Line 103-107: These 5 lines form the basis for your hypothesis and rationale for a major part of the study. Is it possible to elaborate slightly more on the link (and underlying mechanism) between cognitive function in youth and risk for hypertension and cardiovascular diseases?

B. Method:

- Line 140-142: Is it possible to give a bit more context of the terms "fluid cognitive ability" and "crystallised ability"? As a non-expert in this field, but with interest in cardiovascular disease and cognition, this context would be welcome.
- Line 164-166: I advise the authors to include (already in this part of the paper) an explanation why they adjusted for antihypertensive treatment.
- Line 164-166: Some drugs do have blood lowering properties, but are not classified or recognized as anti-hypertensives. Is it possible to give insight which type of drugs the authors defined as "anti-hypertensive medications"? Or how they were recorded in the survey?
- Line 176: It is advisable to give more insight in the term cardiovascular disease. What is exactly meant with the term? Is it possible to describe in subgroups such as myocardial infarct, peripheral artery disease, heart failure with reduced ejection fraction, etc. How was it asked in the survey?
- Line 176: is it a possibility to add 'history of diabetes mellitus'? It seems to be missing in table 1.
- Line 186-187: it is not completely clear why the authors used "latent and measured" variables. Systolic/diastolic blood pressure values and cognition test results are directly measurable. Please describe more in detail why certain techniques were used.

C. Results:

- Overall comment: well written and with detailed description of the used statistical techniques, but because of these details the result section is quite overwhelming.
- Line 313-318: this is an interpretation of the results and forms of part of the discussion section.
- Line 334-335: idem, interpretation of results.

D. Discussion:

- Overall comment: well written, honest discussion and to the point.
- Line 345-354: Does a hypothesis still stand when inclusion of covariates disrupts the robustness of a model?
- Line 361-373: the postulated hypotheses are in line with results of certain important clinical trials in the field of hypertension. It could be interesting to put these results in the context of the paper under review.

	□ The SPRINT MIND Investigators for the SPRINT Research Group. Effect of Intensive vs Standard Blood Pressure Control on Probable Dementia: A Randomized Clinical Trial. JAMA. 2019;321(6):553–561. doi:10.1001/jama.2018.21442 □ Peters, R., Beckett, N., Forette, F., Tuomilehto, J., Clarke, R., Ritchie, C., ... & Comsa, M. (2008). Incident dementia and blood pressure lowering in the Hypertension in the Very Elderly Trial cognitive function assessment (HYVET-COG): a double-blind, placebo controlled trial. The Lancet Neurology, 7(8), 683-689.
--	--

REVIEWER	Daniel Nation University of California Irvine
REVIEW RETURNED	17-Mar-2020

GENERAL COMMENTS	The authors used latent growth curve models to investigate associations between blood pressure and cognition in 1,091 individuals born in 1936. The statistical models are elegant and well-designed, including time-varying and time-insensitive covariates as appropriate, and utilizing multiple BP and cognitive measurements. The authors also conducted thoughtful post-hoc analyses to examine the role of medications in these associations. I have some comments to improve the clarity of the manuscript.  1. In the Abstract conclusions, authors state “Our findings indicate an association between early-life cognitive function and later-life blood pressure”. For clarity, please specify the direction of the association. 2. On page 6, lines 89-90, after describing the relationship between hypertension and cognition in older adults, the authors state: “However, there is also evidence for the relationship operating in the opposite direction.” This is confusing as they then go on to state the relationship that has been observed in youth, an entirely different group. I would recommend dropping this sentence entirely, and stating the different relationships observed first in older adults and then in youth. 3. The links between having hypertension and lower cognitive function are not the same as the links between risks of developing hypertension and experiencing stroke/vascular events. These are 4 conceptually different things, and the authors are advised not to compare them as though they were the same. Greater precision (e.g. explaining the link between vascular events and cognition) would improve the quality of the manuscript. 4. Please provide a brief explanation of “gene dosage from sex chromosomes” (page 5, line 100). 5. Please clarify the meaning of this sentence on page 6, lines 108-109: “... we tested two hypotheses regarding the relationships between cognitive functions, some of which steadily decline in mean level in older participants..” 6. Could the authors please provide a brief (1-2 sentence) explanation for how the tests were chosen to represent each subdomain? I question the inclusion of Letter-Number Sequencing and Digit Span Backwards as tests of Memory; these tests have
---

	very different encoding and recall demands compared to Verbal Paired Associates and Logical Memory. 7. Could the authors please clarify how “Blood pressure score” on Figure 1 relates to mmHg measurements? 8. Please indicate directly in the Methods section which model was Model 1A. Model 1A is referenced in the Results, Figure 2, and Supplementary Table 2, but not Methods. 9. On page 13, lines 270-277, please specify for all results that cognitive function being referred to was at age 11. It becomes confusing to read “higher cognitive function men” when cognitive function being referenced was for age 11 (when they were boys). 10. On page 14, lines 335-338, the authors state “being diagnosed with hypertension did not have any protective associations with cognitive function.” I would rephrase this, as hypertension is not generally expected to be a protective factor with regard to cognition to begin with.
--	---

VERSION 1 – AUTHOR RESPONSE

Reviewer(s)' Comments to Author:

Reviewer: 1

Reviewer Name: Rudi Westendorp

Institution and Country: University of Copenhagen, Denmark

Please state any competing interests or state ‘None declared’: none

Please leave your comments for the authors below

This is scholarly analysis of the relation between blood pressure and cognitive function across the eighth decade within the phenotypically rich Lothian Birth Cohort of 1936. Outcomes of this type of analyses are quintessential for a better understanding of cognitive decline in old age which is of interest for a great readership.

Response: We thank the reviewer for his thoughtful and constructive comments, and appreciate his kind words.

The authors go to great length to understand why there were no associations between cognitive function and blood pressure in the eighth decade, and therefore examined the role of medication in post-hoc analyses. In the main analysis (line 160), adjustments for anti-hypertensive medications were made to both systolic and diastolic blood pressure. 15 points were added to systolic pressure and 10 points were added to diastolic pressure if an individual was recorded as taking antihypertensive medication, such as atenolol, lisinopril, bisoprolol, etc. at that wave. I follow this reasoning when testing the hypothesis that the association between higher cognitive function at age 11 years and lower blood pressure in the eighth decade of life is stronger in women than it is in men, as in that case participants would have a ‘falsely’ low recorded blood pressure. However, when the hypothesis is tested that cognitive function and blood pressure will be reciprocally associated with each other across the eighth decade, an etiologically question, statistically adding blood pressure in a prospective design when the real blood pressure is measured, is not obvious.

Response: The aim of this analysis (and all analyses in the bivariate growth curve framework we employed) was to analyze blood pressure constructs that are more representative of “natural” blood pressure, hence our use of adjusted blood pressure variables. If we understand the reviewer’s point

regarding the reciprocal analyses during the 8th decade, measured blood pressure levels might impact cognitive function (and vice versa). We recognize this issue, and part of the purpose of our post-hoc ANOVAs was to take wave, unadjusted blood pressure scores, and medication into account to address questions of cause and effect between blood pressure, medication, and cognitive functions.

In creating this revision, we have since tried to refit our growth curve models with unadjusted blood pressure variables. Unfortunately, our attempts at these models would not converge – replacing adjusted with unadjusted blood pressure causes issues with model fit that we could not overcome. Our best assessment of why comes down to the difficulty in modelling discontinuities when an individual begins taking an anti-hypertensive medication. Blood pressure readings appear to drop suddenly from one wave to the next, which interferes with getting good linear or curvilinear fit to the data. We regret that we were not able to complete these analyses, though we have added mention of this problem to our revised manuscript (lines 310 – 315). We have attempted several additional analyses in place of this in an attempt to get to the root of the question – please see our response to the reviewer’s final comment.

When fitting their posthoc models, there was an inverse main effect of being diagnosed with hypertension at any wave and overall cognitive function level (line 330) suggesting that hypertension may be associated with negative cognitive change between childhood and older age, which is in line with current reasoning on the relation between longstanding hypertension and cognitive decline. However, the following reasoning ‘that being diagnosed with hypertension did not have any protective associations with cognitive function across the eighth decade: it did not improve cognitive function or make up for the negative association between blood pressure and cognitive decline.’ (line 335) is rather ambiguous. Why should a history of hypertension protect against cognitive decline in old age? Please note that this question is different from the pathophysiological reasoning that higher blood pressure in old age may protect against cognitive decline, especially in those with cerebral vascular damage (due to long-lasting hypertension).

Response: In this sample, being diagnosed with hypertension almost invariably means that an individual will be given anti-hypertensive medication. This statement was meant to refer to the impact of medication that results from a diagnosis. In response to this comment and those from the other reviewers, we have clarified this in revised Results sections to explain the hypertension-medication connection.

My first suggestion is that in the second paragraph of the introduction (Line 86) the authors describe a third type of association between blood pressure and cognition, based on pathophysiological reasoning, for instance see <https://www.ncbi.nlm.nih.gov/pubmed/25730401>. From this text, ‘It has been shown that the degree of vascular damage in the systemic and cerebral circulation is linked with lower cerebral blood flow. Long-lasting cerebral hypoperfusion results in neuronal energy crisis and cell death. At the same time, damage of the brain can lead to dysregulation of blood pressure and a further decline in cerebral blood flow. Therefore, what is considered a normal or low blood pressure in individuals with cognitive impairment may not necessarily mean a well-controlled blood pressure.’

Response: We thank the reviewer for this suggestion and the helpful supporting information. We have added material on this to the introduction (lines 94 to 98).

My second suggestion is to study the effect of blood pressure and cognition later in life, making full use of the follow-up design, as blood pressure is a causal determinant of cognition. For an example see <https://www.ncbi.nlm.nih.gov/pubmed/19453303>. In contrast, the structural equation modelling that is being used when analyzing the Lothian Birth cohort, effectively makes it a cross sectional design. The outcomes of cross-sectional analyses on blood pressure and cognition generally indicate a positive association, suggesting that high blood pressure is protective. This is, as a general conclusion unjust and all kinds of biases, confounding and selection mechanisms are at play.

Response: The effect described in Euser et al. (2009) might be in concordance with what we have observed. Through our longitudinal SEM models, we fit linear and quadratic factors of blood pressure change over time, a distinct advantage of this framework, that cannot be directly approached with cross-sectional analyses. The quadratic effect we found in blood pressure across the 8th decade indicates that blood pressure rises on average and then decreases with age. There is greater risk of high blood pressure in the early part of the decade, and greater risk of low blood pressure in later parts of the decade. Following the cognitive epidemiological hypothesis that cognitive functions are associated with better health, higher cognitive overall/lifetime function ought to be associated with better blood pressure regulation at all points in the 8th decade. However, we acknowledge that it is not obvious how cognitive change over the 8th decade ought to be associated with linear or quadratic change in blood pressure, so we have added cross-sectional analyses – again, please see the next reply.

My third suggestion is to combine these two thoughts, i.e. allowing that there is an age and or state (i.e. a history of hypertension and or cerebral vessel disease) interaction between blood pressure and cognition. As a beginning, it may be worthwhile to split the sample in two, analysing the first half of the observation period separate from the second half. There is a strong lead in the data that there is an age and or state interaction at play as ‘Both groups show a quadratic effect, blood pressure rose between ages 70 and 76, and then declined between age 76-79 (line 249).

Response: We thank the reviewer for this suggestion, though we hesitate to break our sample into two sets of two as this would not allow us to identify a reliable baseline or trend, as these require three or more measurements. However, we see the merit of the suggested approach, and have elected to instead provide wave by wave cross-sectional analyses of the same variables, thus allowing for direct wave by wave comparisons of the associations between blood pressure score and the cognitive functions. We have done this with and without hypertension adjustment, and with the general and domain scores of cognitive function (lines 319 – 328, Figure 4, supplementary table 4).

Rudi Westendorp, Copenhagen.

Reviewer: 2

Reviewer Name: Jacobijn Gussekloo

Institution and Country: leiden university medical center

Please state any competing interests or state ‘None declared’: none

Please leave your comments for the authors below

This manuscript is well written and is focused on blood pressure and cognitive function, a very interesting and actual scientific theme. However, since 1) the clinical hypothesis is not very well worked out and 2) the statistical methods and results are very difficult and extensive, I believe this manuscript fits better in a statistical journal instead of this journal for clinical doctors. In detail, I have the following extra points.

A. Introduction:

- Line 77: “...linked to cardiovascular diseases such as coronary artery disease (CAD) and stroke.” I would suggest to add also heart failure because the link between chronic high blood pressure and heart failure is of great clinical importance (the prevalence of heart failure rises to 10% and more among people 70 years and older).

Response: We have added heart failure to this list of circulatory conditions.

- Line 84: “Worldwide, hypertension, age related cognitive decline, and dementia are on the rise.” Important fact, but it pops a little bit out of the blue at the end of the first paragraph of the introduction. Can it be streamlined into the paragraph?

Response: We have rewritten this part of the introduction to try to better streamline the text.

- Line 88: “In some samples of older people hypertension is associated with lower cognitive functioning and faster decline.” I would advise to include ‘the timing’ of hypertension. Do the authors mean hypertension since middle-age (or even adulthood) or development of hypertension at an older age? In pathophysiological context this can be somewhat different.

Response: The evidence cited here, which spans reviews, is understandably mixed, so it is difficult to make the statement more specific. In general, hypertension is a disorder of middle age, and cognitive decline is more extensively studied in older age, and the research reflects these etiologies, i.e. more examinations have looked at cognitive decline that comes after hypertension diagnosis. We have thus rewritten this sentence to read ‘having hypertension is associated with lower cognitive functioning and faster decline’ to reflect this.

- Line 103-107: These 5 lines form the basis for your hypothesis and rationale for a major part of the study. Is it possible to elaborate slightly more on the link (and underlying mechanism) between cognitive function in youth and risk for hypertension and cardiovascular diseases?

Response: In response to this comment and a comment from reviewer 1, we have added more information on the hypothetical links between cognitive function and hypertension.

B. Method:

- Line 140-142: Is it possible to give a bit more context of the terms “fluid cognitive ability” and “crystallised ability”? As a non-expert in this field, but with interest in cardiovascular disease and cognition, this context would be welcome.

Response: Certainly, we have elaborated more on these concepts in the revised text.

- Line 164-166: I advise the authors to include (already in this part of the paper) an explanation why they adjusted for antihypertensive treatment.

Response: We have added more on this issue, in response to this comment, and others from reviewer 1.

- Line 164-166: Some drugs do have blood lowering properties, but are not classified or recognized as anti-hypertensives. Is it possible to give insight which type of drugs the authors defined as “anti-hypertensive medications”? Or how they were recorded in the survey?

Response: The second author examined a full list of medications for each individual at each wave, and labeled that individual as taking anti-hypertensive medication if they were taking any medications with anti-hypertensive properties. Unfortunately, we cannot provide a list of the exact medications that were including in this category because the second author has since passed away.

- Line 176: It is advisable to give more insight in the term cardiovascular disease. What is exactly meant with the term? Is it possible to describe in subgroups such as myocardial infarct, peripheral artery disease, heart failure with reduced ejection fraction, etc. How was it asked in the survey?

Response: The questions asked of participants was “Have you ever had a heart attack, angina, heart valve problem, abnormal heart rhythm or any other heart problem?” Unfortunately, we were not able to break down the condition any further than this. We do not have specific data on heart failure in this sample largely because it is quite rare in the sample; people who suffer from heart failure are usually too frail to come in for testing and drop out of the cohort. We thus do not record heart failure data unless the participant offers the information, which has its own issues with selective report bias.

- Line 176: is it a possibility to add ‘history of diabetes mellitus’? It seems to be missing in table 1.

Response: This has been added to Table 1.

- Line 186-187: it is not completely clear why the authors used “latent and measured” variables. Systolic/diastolic blood pressure values and cognition test results are directly measurable. Please describe more in detail why certain techniques were used.

Response: Latent variables have several advantages over directly measured variables, particularly in the context of growth curve models, which we used here. We have elaborated on the specific advantages of these methods in the revised text (lines 200 to 205).

C. Results:

- Overall comment: well written and with detailed description of the used statistical techniques, but because of these details the result section is quite overwhelming.
- Line 313-318: this is an interpretation of the results and forms of part of the discussion section.
- Line 334-335: idem, interpretation of results.

Response: We have stripped out such sections from the Results, and have aimed to keep these interpretive statements in the Discussion.

D. Discussion:

- Overall comment: well written, honest discussion and to the point.
- Line 345-354: Does a hypothesis still stand when inclusion of covariates disrupts the robustness of a model?

Response: The language we used was ‘This finding provides mixed support for our first hypothesis and the existing supporting literature, as the effect was not robust to the inclusion of covariates.’ This is the most accurate presentation of our findings, and we believe the best presentation for the opening of the discussion, as we acknowledge the limited support for the hypothesis up front, and point to the covariates that could explain why.

- Line 361-373: the postulated hypotheses are in line with results of certain important clinical trials in the field of hypertension. It could be interesting to put these results in the context of the paper under review.

The SPRINT MIND Investigators for the SPRINT Research Group. Effect of Intensive vs Standard Blood Pressure Control on Probable Dementia: A Randomized Clinical Trial. *JAMA*. 2019;321(6):553–561. doi:10.1001/jama.2018.21442

Peters, R., Beckett, N., Forette, F., Tuomilehto, J., Clarke, R., Ritchie, C., ... & Comsa, M. (2008). Incident dementia and blood pressure lowering in the Hypertension in the Very Elderly Trial cognitive function assessment (HYVET-COG): a double-blind, placebo controlled trial. *The Lancet Neurology*, 7(8), 683-689.

Response: We thank the reviewer for raising these papers to our attention. We have added additional discussion comparing our findings to these trials.

Reviewer: 3

Reviewer Name: Daniel Nation

Institution and Country: University of California Irvine

Please state any competing interests or state ‘None declared’: None

Please leave your comments for the authors below

The authors used latent growth curve models to investigate associations between blood pressure and cognition in 1,091 individuals born in 1936. The statistical models are elegant and well-designed, including time-varying and time-insensitive covariates as appropriate, and utilizing multiple BP and cognitive measurements. The authors also conducted thoughtful post-hoc analyses to examine the

role of medications in these associations. I have some comments to improve the clarity of the manuscript.

1. In the Abstract conclusions, authors state “Our findings indicate an association between early-life cognitive function and later-life blood pressure”. For clarity, please specify the direction of the association.

Response: We have done this in the revised manuscript.

2. On page 6, lines 89-90, after describing the relationship between hypertension and cognition in older adults, the authors state: “However, there is also evidence for the relationship operating in the opposite direction.” This is confusing as they then go on to state the relationship that has been observed in youth, an entirely different group. I would recommend dropping this sentence entirely, and stating the different relationships observed first in older adults and then in youth.

Response: Whilst incorporating suggestions from the other reviewers, we have also taken this comment into account, and the revised version of this intro paragraph presents this more along information along the lines suggested, with considerable additional framing.

3. The links between having hypertension and lower cognitive function are not the same as the links between risks of developing hypertension and experiencing stroke/vascular events. These are 4 conceptually different things, and the authors are advised not to compare them as though they were the same. Greater precision (e.g. explaining the link between vascular events and cognition) would improve the quality of the manuscript.

Response: Please see the revised text in the intro (lines 88 to 98), which also aims to respond to the comment above and comments from the other reviews as well. In the next text we have provided some real world context that links vascular physiology to cognition, in order to describe a one of the potential link between these two systems with more precision.

4. Please provide a brief explanation of “gene dosage from sex chromosomes” (page 5, line 100).

Response: We have added an explanation in the revision.

5. Please clarify the meaning of this sentence on page 6, lines 108-109: “... we tested two hypotheses regarding the relationships between cognitive functions, some of which steadily decline in mean level in older participants..”

Response: We have rewritten the beginning of this paragraph to give greater clarity.

6. Could the authors please provide a brief (1-2 sentence) explanation for how the tests were chosen to represent each subdomain? I question the inclusion of Letter-Number Sequencing and Digit Span Backwards as tests of Memory; these tests have very different encoding and recall demands compared to Verbal Paired Associates and Logical Memory.

Response: We have added additional text on this at the end of the Methods section on Cognitive functions (lines 156 to 161). The tests were chosen when the cohort was initiated, over a decade ago. Membership of any test within a domain was determined empirically, in gradually evolving models with very good fit in confirmatory factor analyses. The particular memory tests in question were selected to encompass a broad memory factor and in earlier work, these tests have all been shown to validly contribute to a single cognitive factor for memory.

7. Could the authors please clarify how “Blood pressure score” on Figure 1 relates to mmHg measurements?

Response: Blood pressure score is a unitless construct and is best thought of in terms of sample means and standard deviations. A score of 0 indicates the individual is in line with average blood pressure, and 1 would indicate that an individual was 1SD higher blood pressure at that time. There is no direct relationship to mmHg, and there cannot be because this score incorporates weighted

information from both systolic and diastolic blood pressure. Text to this effect has been added to Figure 1.

8. Please indicate directly in the Methods section which model was Model 1A. Model 1A is referenced in the Results, Figure 2, and Supplementary Table 2, but not Methods.

Response: Thank you for pointing this out. We have clarified in the Methods.

9. On page 13, lines 270-277, please specify for all results that cognitive function being referred to was at age 11. It becomes confusing to read “higher cognitive function men” when cognitive function being referenced was for age 11 (when they were boys).

Response: We have clarified this in the revised text, so that it is clearer that we are referring to cognitive function measures from age 11.

10. On page 14, lines 335-338, the authors state “being diagnosed with hypertension did not have any protective associations with cognitive function.” I would rephrase this, as hypertension is not generally expected to be a protective factor with regard to cognition to begin with.

Response: We have rewritten this sentence, and largely removed the phrase in question.

VERSION 2 – REVIEW

REVIEWER	rudi westendorp Rudi Westendorp Professor of Medicine at Old Age Department of Public Health University of Copenhagen
REVIEW RETURNED	17-Jun-2020

GENERAL COMMENTS	The authors have sufficiently addressed the comments that I raised.
---

REVIEWER	jacobijn gussekkoo LUMC, Leiden, the Netherlands
REVIEW RETURNED	13-Jun-2020

GENERAL COMMENTS	Although the manuscript still has a high statistical content, I am overall more positive:  - Well written (conform first review) - Detailed and stepwise description of complex statistical analyses (conform first review) - Enhanced pathophysiological details and clinical context in the introduction - The authors complied with the feedback to clarify terminology Feedback We sincerely appreciate the fact the authors in the introduction, to some extent, elaborated on the multifaceted relationship between blood pressure, cognitive decline and risk on cardiovascular disease in a clinical context. Hypertension in youth, middle-aged and the oldest old, all have different underlying physiopathological mechanisms and need to be considered as separate slightly
--

	different clinical conditions. Furthermore, clarifying the terms “fluid cognitive ability” and “crystallized ability” and “latent variables”, improves the overall readability of the manuscript. Mentioning the adjustment for anti-hypertensive medication in the methodology section enhances the understanding of the underlying assumptions, however, it is a debatable correction to perform. Therefore, it is a positive aspect that the authors also attempted to fit bivariate LGCMs of unadjusted (for treatment) blood pressure and cognitive function (unfortunately without resulting in additional insights) and added an examination of cross-sectional correlations. The fact that the authors rephrased certain statements in the discussion section and made some postulations more modest in context of the mechanisms of hypertension in old age, enhances the overall validity of the conclusions of this study. Final conclusion: Overall, the storyline, background and readability of the paper is improved, but question for the editor is whether or not this extensive statistical methodology which requires very active reading, fits the scope of the journal.
--	--